# Quantifying the Deficits of Body Water and Monovalent Cations in Hyperglycemic Emergencies

**DOI:** 10.3390/jcm14010025

**Published:** 2024-12-25

**Authors:** Brent Wagner, Mark L. Unruh, Susie Q. Lew, Maria-Eleni Roumelioti, Ramin Sam, Christos P. Argyropoulos, Richard I. Dorin, Todd S. Ing, Mark Rohrscheib, Antonios H. Tzamaloukas

**Affiliations:** 1Division of Nephrology, Department of Medicine, University of New Mexico School of Medicine, Albuquerque, NM 87122, USA; brwagner@salud.unm.edu (B.W.); mroumelioti@salud.unm.edu (M.-E.R.); cargyropoulos@salud.unm.edu (C.P.A.); mrohrscheib@salud.unm.edu (M.R.); 2Kidney Institute of New Mexico, University of New Mexico Health Sciences Center, Albuquerque, NM 87122, USA; 3Raymond G. Murphy Veterans Affairs Medical Center, Albuquerque, NM 87108, USA; mlunruh@salud.unm.edu; 4Department of Medicine, University of New Mexico School of Medicine, Albuquerque, NM 87122, USA; 5Division of Nephrology, Department of Medicine, School of Medicine and Health Sciences, George Washington University, Washington, DC 20037, USA; sqlew@gwu.edu; 6Division of Nephrology, Department of Medicine, Zuckerberg San Francisco General Hospital, University of California in San Francisco School of Medicine, San Francisco, CA 94110, USA; ramin.sam@ucsf.edu; 7Division of Endocrinology, Department of Medicine, Raymond G. Murphy Veterans Affairs Medical Center, University of New Mexico School of Medicine, Albuquerque, NM 87108, USA; rdorin@salud.unm.edu; 8Department of Medicine, Stritch School of Medicine, Loyola University Chicago, Maywood, IL 60153, USA; 9Research Service, Department of Medicine, Raymond G. Murphy Veterans Affairs Medical Center, University of New Mexico School of Medicine, Albuquerque, NM 87108, USA

**Keywords:** diabetic ketoacidosis, hyperglycemic hyperosmolar state, dehydration, hypovolemia

## Abstract

**Background/Objectives:** Hyperglycemic emergencies cause significant losses of body water, sodium, and potassium. This report presents a method for computing the actual losses of water and monovalent cations in these emergencies. **Methods:** We developed formulas for computing the losses of water and monovalent cations as a function of the presenting serum sodium and glucose levels, the sum of the concentrations of sodium plus potassium in the lost fluids, and body water at the time of hyperglycemia presentation as measured by bioimpedance or in the initial euglycemic state as estimated by anthropometric formulas. The formulas for computing the losses from hyperglycemia were tested in examples of hyperglycemic episodes. **Results:** The formulas were tested in two patient groups, those with or without known weight loss during the development of hyperglycemia. In the first group, these formulas were applied to estimate the losses of body water and monovalent cations in (a) a previously published case of a boy with diabetic ketoacidosis and known weight loss who, during treatment not addressing his water deficit, developed severe hypernatremia and (b) a comparison of water loss computed by this new method with the reported average fluid gained during treatment of the hyperglycemic hyperosmolar state in a published study. In the second group, the formulas were applied in hypothetical subjects with varying levels of initial body water, serum sodium, and glucose at the time of hyperglycemia and sums of sodium and potassium concentrations in the lost fluids. **Conclusions:** Losses of body water and monovalent cations, which determine the severity of dehydration and hypovolemia, vary significantly between patients with hyperglycemic emergencies presenting with the same serum glucose and sodium concentrations. These losses can be calculated using estimated or measured body water values. Prospective studies are needed to test this proof-of-concept report.

## 1. Introduction

Hyperglycemic emergencies, including diabetic ketoacidosis (DKA), the nonketotic hyperglycemic hyperosmolar state (HHS), and combined DKA-HHS, are associated with severe clinical manifestations and mortality [1,2,3,4,5,6,7]. Hyperglycemia may also worsen the outcome of other clinical conditions [8,9]. Hyperglycemic episodes may cause profound deficits in body water and extracellular volume. Dehydration is a deficit of body water relative to the amount of osmotically effective solutes in body fluids, regardless of whether the amount of body solutes is low, in the normal range, or high. Dehydration causes hypertonicity, expressed as an increased concentration of the effective solutes in body fluids. Hypovolemia is caused by the loss of sodium salts, mainly sodium chloride, accompanied by fluid loss. The distinction between dehydration and hypovolemia in hyperglycemia, which guides the fluid management of hyperglycemic emergencies, has been stressed in the literature [10]. Neurological dysfunction and acute renal failure constitute two severe clinical consequences of hyperglycemia resulting from dehydration or hypovolemia.

The neurological manifestations resulting from hyperglycemia, e.g., coma or seizures, are related to hyperglycemic hypertonicity [11,12]. Hypertonicity causes osmotic transfer of fluid from the intracellular into the extracellular compartment. In hyperglycemia, hypertonicity is the consequence of two pathophysiologic processes: (a) Gain in extracellular solute (glucose) [13,14]. Hypertonicity as an exclusive consequence of glucose gain is associated with hyponatremia resulting from the dilution of the extracellular fluid caused by the osmotic exit of fluid from the intracellular compartment [11,12,13,14,15,16,17]. (b) Loss of body water in relative excess to the losses of body sodium and potassium (dehydration) through osmotic diuresis [15,18,19,20]. As a result of the hyperglycemic osmotic diuresis, the serum sodium concentration ([*Na*]*_S_*) is higher than the expected value from glucose gain [21]. [*Na*]*_S_* values in the range of hypernatremia have been reported in patients with HHS or DKA-HHS [22]. The biochemical indices of tonicity include [*Na*]*_S_*, serum sodium corrected for the degree of hyperglycemia without any changes in body water or monovalent cations ([*Na*]*_Cor_*), and the tonicity formula (see below) [23]. An [*Na*]*_Cor_* value in the hypernatremic range indicates the presence and the degree of dehydration, which develops from loss of water more than any loss of effective solute in the body fluids [21,22].

Acute renal failure in hyperglycemia is the consequence of the loss of effective arterial blood volume produced by severe hypovolemia [24]. When hyperglycemic emergencies develop in patients with preserved renal function, the loss of extracellular volume through osmotic diuresis constitutes, by far, the major mechanism for the development of hypovolemia [22].

The correction of dehydration and hypovolemia constitutes a significant part of the treatment of hyperglycemic syndromes [1,2,3,4,5,6,11,25]. Treatment guidelines contain average estimates of fractional or actual losses of body water which should guide the selection of the volume of the replacement solutions [1,2,3,4,5,6,26] and provide schemes for the composition and the rate of fluid replacement. Clinical estimates of body water and extracellular volume deficits in hyperglycemic emergencies are inaccurate [27,28,29]. A previous report presented formulas for computing the fractional deficits of body water and monovalent cations (sodium plus potassium) at presentation with hyperglycemia [22]. These formulas used the levels of the blood glucose ([*Glu*]*_S_*), the ([*Na*]*_Cor_*), and the sum of the concentrations of sodium plus potassium in the lost fluids ([*Na*]*_Lost_* + [*K*]*_Lost_*). An important finding in this report is that the fractional losses of water and monovalent cations may differ significantly among patients presenting with the same [*Na*]*_S_* and [*Glu*]*_S_* [22]. This finding suggests that the prescribed fluid replacements in hyperglycemia should be individualized and not based on reported average losses. In this report, we present a quantitative method for estimating the actual deficits, not the fractional deficits, of body water and monovalent cations at presentation with hyperglycemia. This method uses the same parameters for computing the actual deficits as the method for computing the fractional deficits plus measured or estimated body water values.

## 2. Materials and Methods

### 2.1. Development of Formulas

The formulas from a previous report compute the fractional deficits of body water and monovalent cations secondary to losses in severe hyperglycemia through osmotic diuresis and through the skin, gastrointestinal, and respiratory systems [22]. The fundamental concept used for the construction of these formulas is that the [*Na*]*_Cor_* is determined by the fraction (total body sodium plus potassium)/total body water the same way that [*Na*]*_S_* is determined by this same fraction in states without an excess of any solute with extracellular distribution, other than sodium salts [30,31].

This report presents formulas that compute the actual deficits of body water and monovalent cations in hyperglycemia. These formulas were based on the same fundamental concept as those computing the fractional losses [22]. Table 1 presents the abbreviations of the parameters used to construct the formulas.

Table 2 presents the formulas for computing the actual losses of body water and monovalent cations and the estimates of total body water in the baseline euglycemic and hyperglycemic states.

Formula (1), obtained from the report by Al-Kudsi and co-investigators [16], provides the calculated [*Na*]*_Cor_* to a [*Glu*]*_S_* of 5.6 mmol/L (100 mg/dL) using Katz’s coefficient of a 1.6 mmol/L increase in [*Na*]*_S_* for each 5.6 mmol/L decrease in [*Glu*]*_S_* [32]. Formula (2) presents the relation between the total volume of body water lost during the development of hyperglycemia (*V_Lost_*) and the total body water values at baseline euglycemia (*TBW*_1_) and hyperglycemia (*TBW*_2_) [23]. Formula (3) computes in units of mmol the actual losses of monovalent cations ({[*Na*]*_Lost_* + [*K*]*_Lost_*} × *V_Lost_*) [22].

Formulas (2) and (3) were used to compute Formulas (4)–(8). The monovalent cation concentrations [*Na*]*_Cor_*, [*Na*]*_S_*_1_, [*Na*]*_Lost_*, and [*K*]*_Lost_* (Table 1) were entered in all four Formulas (4)–(8). Formula (4) computes the *V_Lost_* using the *TBW*_1_ and the four monovalent cation concentrations in formula (3). Formula (5) computes the sum [*Na*]*_Lost_* + [*K*]*_Lost_* as a function of *TBW*_1_, *V_Lost_*, [*Na*]*_S_*_1_, and [*Na*]*_Cor_*. Formula (6) computes the *TBW*_2_ as a function of *TBW*_1_ and the monovalent cation concentrations. Formula (7) computes *TBW*_1_ using the *V_Lost_* and the monovalent cation concentrations. Formula (8) computes *TBW*_1_ as a function of *TBW*_2_. Note that each of the Formulas (4)–(8) requires the estimate or measurement of one or two of the water volumes *TBW*_1_, *TBW*_2_, and *V_Lost_*.

### 2.2. Computation of Losses: Information Needed

The clinical inquiries required for managing the losses of water and electrolytes in a hyperglycemic emergency include the following: (a)Identification of the condition that precipitated this emergency. Most cases of DKA or HHS are precipitated by other clinical conditions [1,2,4,5,6,22,26,33,34,35,36,37,38,39,40,41,42].(b)Identification of the clinical manifestations of the hyperglycemic emergency [1,2,3,4,5,6,7,8,9,10,11,12,13,14,15,16,17,18,19,20,21,22,23,24,25,26]. The manifestations relevant to the management of fluid and monovalent cation losses include the history of fluid loss primarily through osmotic diuresis and through the gastrointestinal tract or the skin [1,2,4,5,20,22,43]. The body weight lost during the development of a hyperglycemic syndrome provides a reasonable estimate of the lost body water. Estimating the water lost is more difficult in patients whose body weight before the hyperglycemic syndrome is unknown.(c)Checking whether any clinical condition, for example congestive heart failure, chronic kidney disease, or liver cirrhosis, exists that will limit the renal excretion of infused fluids and electrolytes. End-stage renal disease (ESRD) limits greatly the renal excretion of fluids and electrolytes. Infusion of saline in patients with ESRD treated for DKA produced severe symptomatic hypervolemia [44]. Physical examination (e.g., significant edema and hypertension), monitoring of urinary output, imaging (e.g., chest X-ray and pulmonary ultrasound), and laboratory tests (e.g., serum creatinine, urea, and liver function tests) should be used to identify the presence of conditions limiting the urinary excretion of fluids in comatose hyperglycemic patients without a known history.(d)Measuring the laboratory values required to determine the level of tonicity. Tonicity, determined by the concentrations of solutes distributed by and large either in the intracellular compartment, e.g., sodium salts, or in the intracellular compartment, e.g., potassium salts [11,23], causes osmotic fluid shifts between the intracellular and extracellular compartment when abnormal [23,45,46]. Solutes distributed in body water, e.g., urea or ethanol, contribute to osmolarity but not tonicity. As noted, the hypertonicity in hyperglycemia is the consequence of two processes, gain of glucose in the extracellular compartment and loss of water in excess of the loss of sodium and potassium through osmotic diuresis in all the patients who have preserved kidney function plus from the gastrointestinal tract or the skin in some cases. It is essential to accurately quantitate these two parts of hypertonicity, because the part caused by glucose gain will be corrected by normalization of [*Glu*]*_S_* while the part caused by external losses is corrected by infusion of solutions with an appropriately calculated volume and degree of hypotonicity.

The management of hyperglycemic emergencies requires several laboratory tests [1,2,3,4,5,26]. The laboratory values that are needed for selecting the tonicity of the replacement solutions to correct dehydration in hyperglycemia include serum osmolality, [*Na*]*_S_*, and [*Glu*]*_S_*. Figure 1 shows a scheme for computing the component of hyperglycemic hypertonicity which is the result of external losses.

The first step in computing the degree of hypertonicity in hyperglycemia consists of ruling in or out pseudohyponatremia, which is encountered in hyperglycemic emergencies with high serum lipid levels when [*Na*]*_S_* is measured by indirect potentiometry [47,48,49]. Pseudohyponatremia in hyperglycemia may be masked by other influences on [*Na*]*_S_*, including the extracellular glucose gain and the loss of hypotonic fluid [50]. Pseudohyponatremia is detected in hyperglycemia by measuring serum osmolality, computing serum osmolarity, and calculating the osmol gap [50]. Values of the osmol gap > 10 mmol/L [51] suggest the presence of pseudohyponatremia, which is confirmed as shown in Figure 1.

The second step in the management of hyperglycemic dehydration consists of computing [*Na*]*_Cor_*. [*Na*]*_Cor_* is the appropriate parameter for establishing the presence and the degree of dehydration in hyperglycemia, while the tonicity formula (2 × [*Na*]*_S_* + [*Glu*]*_S_* in mmol/L) overestimates the degree of dehydration, and [*Na*]*_S_* underestimates this degree [11,21,22,23,52]. In the setting of hyperglycemia, the presence and degree of dehydration is indicated by [*Na*]*_Cor_* in the range of hypernatremia [22]. In this case, the volume of water needed to normalize [*Na*]*_S_* after the correction of hyperglycemia without any changes in body sodium or potassium is calculated by entering [*Na*]*_Lost_* + [*K*]*_lost_* = 0 and an estimate of *TBW*_1_ in Formula (4) (Table 2). The values of water lost computed by Formula (4) at progressively higher sums [*Na*]*_Lost_* + [*K*]_Lost_ and the same *TBW*_1_ are progressively greater than the value computed when this sum is zero.

The following segments on the computation of water and monovalent cation losses in patients with hyperglycemia apply to patients with hypovolemia documented by history (polyuria and thirst), clinical examination (orthostatic hypotension, tachycardia, and low blood pressure), routine laboratory tests (high serum urea and creatinine levels, especially in patients without prior history of chronic kidney disease), and in some instances special tests addressing volume status including sonography, e.g., respiratory variability of inferior vena cava diameter, and special laboratory tests, e.g., N-terminal pro-brain natriuretic peptide [53,54,55]. Fluid infusion in clinically hypovolemic patients with hyperglycemic emergencies and medical conditions limiting the renal excretion of water and electrolytes requires particularly close clinical monitoring and additional measures, e.g., pulmonary ultrasonography [56].

### 2.3. Computation of Losses: Group with Known V_Lost_ (Group A)

Figure 2 provides a scheme for estimating the volume and composition of the replacement solutions in patients with known weight loss during the development of hyperglycemia.

Recent premorbid weight measurements should be compared to admission values when estimating *TBW*_1_ and *TBW*_2_. Obtaining accurate body weights in health care appointments of patients with diabetes would enable the calculation of *TBW*_1_ and *V_Lost_*. An anthropometric formula, for example the Watson formulas for adults [57] or the Mellits–Cheek formulas for children [58], can be used to estimate *TBW*_1_ at the baseline pre-hyperglycemic state. We suggest measuring *TBW*_2_ and extracellular volume by bioimpedance at presentation with hyperglycemia. Bioimpedance measurements, which are used to monitor the body fluid status of patients who require the management of their body fluids [59], have been applied extensively to monitor fluid status during hemodialysis [60]. In patients on dialysis, bioimpedance has medium to high accuracy, high reproducibility, difficulties in some patient categories, and is both not available universally and is time- and personnel-intensive [54]. Measurement of *TBW*_2_ by bioimpedance is not needed for estimating the volume of the infusate in hyperglycemic patients with known weight loss but can be used to evaluate the accuracy of bioimpedance in hyperglycemic emergencies. Agreement between the *TBW*_2_ value measured by bioimpedance and the value computed by Formula (6) and agreement between the *TBW*_1_ value from the anthropometric formula and the value calculated by Formula (7) would provide support for the use of bioimpedance measurements in the group in which the weight lost is unknown.

The next step consists of computing [*Na*]*_Cor_* by Formula (1), after evaluating for pseudohyponatremia. When [*Na*]*_Cor_* is in the eunatremic or in the hyponatremic range, the estimated losses represent hypovolemia without dehydration. The prescribed replacement solution in this case is isotonic with a volume equal to the estimated *V_Lost_*. Hypotonic solutions may be needed subsequently if [*Na*]*_Cor_* increases during treatment because of ongoing osmotic diuresis. Substantial numbers of patients have normal, or even low, in rare instances, [*Na*]*_Cor_* values at presentation with DKA [61]. For example, in the large PECARN study of fluid infusion rates in pediatric DKA [62], the average [*Na*]*_Cor_* was 140.8 mmol/L in one group and 140.7 mmol/L in the other group [61].

When [*Na*]*_Cor_* is in the hypernatremic range, the required total volume of the replacement solution, which is equal to the calculated *V_Lost_*, is hypotonic, and its tonicity is computed by Formula (5). The guidelines advocate infusing initially isotonic saline in patients with symptomatic hypovolemia [1,2,3,4,5,6,26]. Isotonic saline infused in patients with hypovolemia and hypertonicity improves the hypovolemia and decreases the degree of hypertonicity [63]. The [*Na*]*_Cor_*, the volume of urine, and the [*Na*]*_Lost_* and [*K*]*_Lost_* in the urine should be monitored during treatment and should be used to calculate the volume and composition of the hypotonic solutions which will be infused after the infusion of the isotonic saline in patients with [*Na*]*_Cor_* in the hypernatremic range at presentation.

### 2.4. Computation of Losses: Group with an Unknown V_Lost_ (Group B)

Figure 3 provides a scheme for estimating the volume and tonicity of the replacement solutions in this group of hyperglycemic patients.

Patients with [*Na*]*_Cor_* in the eunatremic or hyponatremic range and hypovolemia diagnosed by clinical and laboratory features should initially receive infusions of isotonic solutions. The volume of the infusate should be determined by monitoring the clinical and laboratory features of hypovolemia during the treatment. Like the corresponding patient subgroup in Group A, patients in Group B presenting with low or normal [*Na*]*_Cor_* may require infusion of hypotonic solutions later because of excessive water loss through osmotic diuresis during the treatment.

The management of fluid and monovalent cation losses in patients with [*Na*]*_Cor_* in the hypernatremic range requires knowledge of at least one of the volumes *TBW*_1_, *TBW*_2_, or *V_Lost_* plus an estimate of the sum [*Na*]*_Lost_* + [*K*]*_Lost_*. *TBW*_2_ is the only volume that can be assessed at presentation. Bioimpedance provides a suitable method for measuring *TBW*_2_. Dehydration is present in relatively few patients with no clinical or laboratory evidence of hypovolemia and [*Na*]*_Cor_* in the hypernatremic range [22]. Our suggested approach to initial estimation of the volume of water required to correct the dehydration in such patients is by calculation of the *TBW*_1_ by Formula (8) using the *TBW*_2_ value obtained by bioimpedance, and with [*Na*]*_Lost_* + [*K*]*_Lost_* = 0 in its denominator. *V_Lost_* can then be calculated by Formula (2).

The literature has provided evidence that the patients with HHS or DKA-HHS routinely have both dehydration and hypovolemia. Computing *TBW*_1_ and *V_Lost_* using Formulas (8) and (2), respectively, in patients with dehydration and hypovolemia, unknown weight loss, and *TBW*_2_ measured by bioimpedance, or any other method, requires inserting estimates of the sum [*Na*]*_Lost_* + [*K*]*_Lost_* in Formula (8). A review of studies measuring monovalent cation concentrations in the urine of subjects with hyperglycemic osmotic diuresis reported a range between 60 and 110 mmol/L for the average values of the sum [*Na*]*_Lost_* + [*K*]*_Lost_* [21]. We suggest measuring the urine sodium and potassium concentrations at presentation. *TBW*_1_ computed by Formula (8) and the volume and composition of the replacement solution equal to the *V_Lost_* computed by Formula (4) with a sum of monovalent cation concentrations of 60 mmol/L in the lost fluid should be prescribed for the infusate initially [22]. It is probable that a higher volume of infusate with a higher sum of monovalent cation concentrations will be needed when the sum of urinary sodium and potassium concentrations at presentation is higher than 60 mmol/L. Monitoring during the treatment period for clinical and laboratory indicators of hypovolemia, serum tonicity indices, repeated measurements of body water by bioimpedance, urine volume and urinary sodium and potassium concentrations, and repeated calculations of *V_Lost_* and its composition by Formulas (8) and (2) using the relevant measurements is required.

Two issues should be stressed during treatment of any hyperglycemic emergency: (a)The computed values of *V_Lost_* and its monovalent cation composition represent the volume and composition of the infusate that should be retained in the body. It is imperative to quantitate the losses of fluid and electrolytes during treatment, which occur through osmotic diuresis in most cases, and to replace the part of the infusate that is lost.(b)The replacement of potassium losses is guided by repeated measurements of the serum potassium concentration. The sodium concentration of the infusate should be reduced when potassium salts are also infused by adding sterile water [64] or dextrose solution so that the sum of sodium plus potassium concentrations in the infusate is equal to the calculated sum by Formula (5). The guidelines recommend dextrose addition to the infused solutions during treatment of hyperglycemic emergencies when [*Glu*]*_S_* reaches 11.2 mmol/L (200 mg/dL) in DKA or 16.8 mmol/L (300 md/dL) in HHS [2] or 14 mmol/L (252 mg/dL) in HHS [26]. Dextrose solution infusion at higher [*Glu*]*_S_* levels can be performed when a potassium salt is added to the hypotonic saline.

## 3. Results

There are no patients with hyperglycemia treated by the method presented in this report. The examples from the literature of patients with hyperglycemia treated by other methods (group A) illustrate the importance of addressing both dehydration and hypovolemia, and the results of the hypothetical patients (group B) illustrate the importance of including the values of [*Na*]*_Cor_*, [*Na*]*_Lost_* + [*K*]*_Lost_*, and *TBW* when calculating the volume and composition of the solutions replacing the losses of hyperglycemic emergencies.

### 3.1. Group A

#### 3.1.1. Importance of Addressing Both Hypovolemia and Dehydration

We present the computed fluid and monovalent cation losses in a reported case of a 14-year-old boy admitted with DKA-HHS, hypotension, and seizures [25]. Table 3 shows the laboratory and computed values relevant to fluid replacement on admission.

On admission, his serum potassium was 2.4 mmol/L, and he had decreased serum total carbon dioxide (TCO_2_) and high serum anion gap, ketone bodies, and levels of urea and creatinine. The infusate should have a volume of 9.8 L and a sum of concentrations of sodium plus potassium of 80 mmol/L. The patient received in this admission 5.4 L of saline with a sodium concentration of 154 mmol/L containing various concentrations of potassium chloride and developed large diuresis as his renal function improved. He developed a deep coma. His [*Na*]*_Cor_* increased from 169.0 to 180.2 mmol/L, and he required transfer to a second hospital, where he had a prolonged hospitalization [25]. The issue with his treatment in the first hospital was that while hypovolemia was corrected, dehydration deteriorated. Neither the necessity for hypotonic infused solutions nor the effect of osmotic diuresis on water balance were addressed. The new method proposed in this report specifically addresses these issues.

#### 3.1.2. Calculations When V_Lost_ and Its Composition Are Known

Comparisons with results obtained by the method presented in this report are feasible in this case. The study of HHS by Arieff and Carroll [43] allows such a comparison. Table 4 shows the relevant values in their study and calculations by the method presented in the present report. Note that the study of Arieff and Carroll allows comparisons for only the mean values of *V_Lost_*. Table 4 shows the values allowing the computation of the mean fractional water loss by the method in the present report.

Using the volume of fluid retained, body weight, and the measured changes in plasma volume, Arieff and Carroll computed mean fractional losses from hyperglycemia equal to 0.253 for the extracellular volume and 0.221 for the intracellular volume [43]. If body fluids were normal at baseline euglycemia and the extracellular volume was 40% of the *TBW*_1_ [65], the fractional body water loss calculated by Arieff and Carroll was 0.253 × 0.4 + 0.221 × 0.6 = 0.234. The close values of the estimates of fractional water loss computed by Arieff and Carroll and by the method in this present study should be interpreted with caution. The estimates of water and monovalent cation retention were performed only in seven episodes in the Arieff and Carroll study, while we used the mean values for [*Glu*]*_S_*_2_ and [*Na*]*_S_*_2_ for all 33 episodes in Table 1 of their report for the calculations by the method presented in this report. In addition, we assumed a normal distribution of body water between the intracellular and extracellular compartments in the baseline euglycemic state.

### 3.2. Group B

The method for assessing fluid and electrolyte losses in Group B patients requires a measurement of *TBW*_2_ along with estimates of the sum [*Na*]*_Lost_* + [*K*]*_Lost_*. Examples of patients having these measurements are currently missing. We will provide hypothetical examples illustrating the importance of different values of *TBW*_1_, of the sum [*Na*]*_Lost_* + [*K*]*_Lost_*, and of [*Na*]*_Cor_* in computing *V_Lost_*. In all the examples, [*Na*]*_S_*_1_ was 140 mmol/L. An example of these calculations is provided for a hypothetical patient with [*Na*]*_Cor_* = 173.6 mmol/L and *TBW*_2_ = 30 L: (a) if [*Na*]*_Lost_* + [*K*]*_Lost_* = 0, *TBW*_1_ = 30/{1 − (173.6 − 140)/173.6} = 37.2 L (Formula (7)), and *V_Lost_* = 37.2 − 30 = 7.2 L (Formula (2)). (b) If [*Na*]*_Lost_* + [*K*]*_Lost_* = 60 mmol/L, *TBW*_1_ = 30/{1 − (173.6 − 140)/(173.6 − 60)} = 42.6 L, and *V_Lost_* = 42.6 − 30 = 12.6 L.

A previous report provided the mean or median values of [*Na*]*_Cor_* in studies of hyperglycemia with [*Na*]*_S_*_2_ in the hypernatremic range [22]. [*Na*]*_Cor_* values from these studies were applied for the estimation of *V_Lost_* in these hypothetical calculations. The mean or median [*Na*]*_S_*_2_ was in the hypernatremic range in four reports [22,33,66,67]. In addition, the mean [*Na*]*_S_*_2_ was in the hypernatremic range in the group of HHS patients in a deep coma in the study of Fulop and co-investigators [68], in the group of deceased patients with HHS in the study of Wachtel and collaborators [69], and in the group of HHS patients who developed rhabdomyolysis in the study of Singhal and co-authors [70]. Table 5 shows the mean or median values of [*Glu*]*_S_*_2_, [*Na*]*_S_*_2_, and [*Na*]*_Cor_* in these seven groups of hyperglycemic patients.

The average [*Na*]*_Cor_* value of the studies in Table 5, weighed for the number of cases in each report, is 167.0 mmol/L. Figure 4 illustrates the contributions of the values of *TBW*_1_ and of the sum [*Na*]*_Lost_* + [*K*]*_Lost_* to the estimates of fluid losses during development of hyperglycemia. This Figure presents estimates of *V_Lost_* obtained by Formula (4) in two hypothetical patients with an [*Na*]*_Cor_* of 167.0 mmol/L at [*Na*]*_Lost_* + [*K*]*_Lost_* sums varying between zero and 110 mmol/L. The first patient was a boy with DKA, a height of 140 cm, and a pre-hyperglycemic weight of 35 kg, in whom the Mellits–Cheek formula [58] computes a *TBW*_1_ of −21.993 + 0.209 × 140 + 0.406 × 35 = 21.48 L. The second patient was a woman with HHS, a height of 170 cm, and a pre-hyperglycemic weight of 90 kg, in whom the Watson formula for women [57] computes a *TBW*_1_ of −2.07 + 0.1609 × 170 + 0.2466 × 90 = 38.27 L.

The contributions of the values of [*Na*]*_Cor_*, in addition to the contributions of *TBW*_1_ and the sum [*Na*]*_Lost_* + [*K*]*_Lost_*, to the estimates of *TBW*_1_ and *V_Lost_* were computed in nine hypothetical patients. Among these patients, four had the highest (173.6 mmol/L) and four had the lowest (157.5 mmol/L) [*Na*]*_Cor_* values in Table 5. For each [*Na*]*_Cor_* value, hypothetical *TBW*_2_ measurements of 20 and 40 L by bioimpedance and two values of [*Na*]*_Lost_* + [*K*]*_Lost_*, zero and 60 mmol/L, were used in the calculations in eight patients. The ninth patient with an [*Na*]*_Cor_* of 157.5 mmol/L and a *TBW*_2_ of 40 L had an [*Na*]*_Lost_* + [*K*]*_Lost_* sum of 110 mmol/L. Table 6 shows the computed values of *TBW*_1_ by Formula (8) and *V_Lost_* by Formula (2), plus the ratio *V_Lost_*/*TBW*_1_ in these patients.

Note that regardless of the size of *TBW*_1_, the fraction *V_Lost_*/*TBW*_1_ is the same in all calculations by Formulas (4), (6), (7), or (8) when the fraction ([*Na*]*_Cor_* − [*Na*]*_S_*_1_)/{[*Na*]*_Cor_* − ([*Na*]*_Lost_* + [*K*]*_Lost_*)} is the same. Also note the big difference in the estimates of *V_Lost_* between the [*Na*]*_Lost_*+ [*K*]*_Lost_* sums of 60 and 110 mmol/L (patients 8 and 9).

## 4. Discussion

Corrections of dehydration and hypovolemia constitute major components of the management of hyperglycemic emergencies. The presented examples using the method presented in this report support the conclusion that the estimation of fluid and electrolyte losses and replacement strategies must be individualized. We suggest that patients with hyperglycemic emergencies should be treated in intensive care units by teams well trained for this purpose using a computer protocol. Availability of point-of-care laboratory tests for measuring glucose, sodium, and potassium concentrations in body fluids is important. Serial measurements by bioimpedance may be difficult in some clinical practices. Also, in the absence of clinical outcomes data using the method suggested herein, it remains to be determined whether calculations by the formulas of this report, repeated assessments of body water and extracellular volume by bioimpedance, and monitoring of urinary electrolyte concentrations during the treatment of severe hyperglycemic episodes yields measurably superior clinical outcomes compared to the current standards of care.

The computation of the losses requires an estimate of the body water. The estimates of the body water deficit, which are computed by entering zero values for the sum [*Na*]*_Lost_* + [*K*]*_Lost_* in Formula (4) for Group A or in Formula (8) for Group B, represent the volume of water that must be gained to correct the dehydration at presentation. This water volume should be computed because it represents the minimal volume of water that is required to correct the hypertonicity of the effective solutes in the body at presentation. The volumes of water needed to correct the water deficit from both dehydration and hypovolemia depend on the sum [*Na*]*_Lost_* + [*K*]*_Lost_* and are always higher than the corresponding volumes from dehydration alone (Figure 4, Table 6).

The lost volumes computed by Formulas (4) or (8) represent volumes and monovalent cation amounts which should be retained in the body. As long as [*Glu*]*_S_* remains elevated during the treatment, osmotic diuresis persists and may even increase in volume as renal function improves with the correction of hypovolemia [25]. The fluid losses during treatment should be accounted for by modifying the infused solutions. Monitoring the clinical status, [*Glu*]*_S_*, [*Na*]*_S_*, [*Na*]*_Cor_*, serum potassium, urine volume, and urinary monovalent cation concentrations during treatment is recommended [1,2,4,6,25,26,71,72,73]. We suggest monitoring body water by bioimpedance and repeated computations of the losses in Group B patients, in whom the first estimates will be obtained using assumed [*Na*]*_Lost_* + [*K*]*_Lost_* sums.

During correction of the hyperglycemia, the extracellular volume of the patients increases through the infused saline and decreases through both the osmotic diuresis and the osmotic transfer of fluid from the extracellular into the intracellular compartment [11]. To correct the hypovolemia, the gain in extracellular volume should exceed the losses. The volumes of saline infused and of osmotic diuresis can be measured. The loss of extracellular volume through osmotic fluid transfer into the intracellular compartment secondary to the decrease in [*Glu*]*_S_* is not negligible. In dialysis patients treated by insulin infusion for hyperglycemia, a method based on the observed change in [*Na*]*_S_* [74] computed an average fractional decrease in extracellular volume of 0.013 L/L per 5.6 mmol/L (100 mg/dL) decrease in [*Glu*]*_S_* [75]. For a [*Glu*]*_S_* of 40.5 mmol/L (729 mg/dL) and an extracellular volume of 20 L, the correction of hyperglycemia to a [*Glu*]*_S_* of 5.6 mmol/L (100 mg/dL) would result in a loss of extracellular volume equal to 0.013 × 20 × ({40.5 − 5.6}/5.6) = 1.62 L. Svensson and co-investigators computed that approximately half of the volume of 0.9% saline infused in hyperglycemic patients with preserved renal function moved intracellularly [76]. Bioimpedance allows repeated measurements of the extracellular volume during treatments resulting in volume changes [60,77]. Monitoring the extracellular volume, along with total body water, during correction of a hyperglycemic emergency would assist in the correction of hypovolemia.

Several sets of guidelines propose the use of isotonic solutions initially to correct dangerous hypovolemia. Isotonic solutions, which decrease the tonicity of body fluids when infused into hyperglycemic patients with [*Na*]*_Cor_* in the hypernatremic range, should be used initially in all episodes of hyperglycemia. Infusion of potassium salts should be guided by serum potassium concentration measurements. Repeated [*Na*]*_Cor_* calculations and serum potassium measurements should guide the composition of the hypotonic solutions infused later during treatment to correct the components of hyperglycemic hypertonicity from losses of fluid and monovalent cations prior to the start of treatment and during treatment [21,45,61,73,78,79].

While symptomatic volume deficits should be corrected rapidly, there are significant neurological risks, including cerebral edema [80,81,82], cerebral herniation [83,84], and osmotic myelinolysis [85,86], attributed to rapid correction of the hyperglycemic hypertonicity and other factors. The rate of correction of hypertonicity during the treatment of hyperglycemia should be slow. One set of guidelines proposed an hourly decline of 2.8–4.2 mmol/L (50–75 mg/dL) in [*Glu*]*_S_* and ≤3 mmol/L in serum tonicity [87]. The rate of decrease in tonicity from the correction of the hyperglycemia calculated using the Katz coefficient [32] is 5.6 − 2 × 1.6 = 2.4 mmol/L per 5.6 mmol/L decrease in [*Glu*]*_S_*. At the rate of decline of [*Glu*]*_S_* proposed by the guidelines [86], the hourly rate of decrease in serum tonicity is 1.2–1.8 mmol/L. [*Na*]*_Cor_*, if it is in the range of hypernatremia, should decrease in parallel with [*Glu*]*_S_*. To achieve a 3 mmol/L hourly rate of decrease in tonicity, the hourly rate of decrease in [*Na*]*_Cor_* that corresponds to the recommended rate of decline in [*Glu*]*_S_* hourly rate is 0.6–0.9 mmol/L [25].

The key limitation of the method proposed in this report is the absence of cases treated by it. Its applicability in clinical practice requires prospective studies. Other limitations of the proposed method include potential inaccuracies of the measurement of body water by bioimpedance or estimated by an anthropometric formula and of the Katz coefficient for the change in [*Na*]*_S_* caused by a change in [*Glu*]*_S_* [22], and not accounting for potential changes in the osmotically active extracellular sodium through interactions with the sodium in the non-osmotically active compartment in the polyanionic proteoglycans of the skin, cartilage, bones, muscle, and endothelial cells [88].

The limitations of bioimpedance measurements of body water and extracellular volume have been analyzed in several reports [65,89,90,91,92,93]. Bioimpedance measurements have advantages. They are easy, have no risks or discomfort for the patients, and can be repeated frequently. Further applications of bioimpedance measurements in patients with abnormalities in body water or extracellular volume are called for [94]. One of the aims of studies based on the method proposed in this report should be the evaluation of the accuracy of the bioimpedance measurements of body water and extracellular volume. The application of other methods of measuring body water in clinical settings, including dual-energy X-ray (DEXA) spectroscopy, air displacement plethysmography, nuclear resonance spectroscopy, and extracellular volume, including DEXA spectroscopy, magnetic resonance imaging, and infusion of markers measuring the glomerular filtration rate [65], could also be studied in hyperglycemic emergencies.

The limitations of the anthropometric formulas estimating body water have been evaluated in patients on dialysis [95,96,97]. There are two potential sources of error in these formulas, the body composition at euvolemia and the presence of hypervolemia. Regarding the body composition, the published formulas fail to include conditions, including the activity level, nutrition, and chronic illnesses that affect the body composition and therefore the body water content [97]. Anthropometric formulas tend to overestimate body water in obese subjects and to underestimate it in lean subjects [96]. Regarding hypervolemia, the anthropometric formulas developed in normal populations systematically underestimate body water in edematous subjects [95]. Formulas developed in populations with fluid excess [96,98] are accurate only for one degree of fluid gain [97].

Regarding the computation of [*Na*]*_Cor_*, coefficients of the change in [*Na*]*_S_* between 1.35 and 4.0 mmol/L per 5.6 mmol/L change in [*Glu*]*_S_* have been reported [79]. The coefficient of 1.6 mmol/L per 5.6 mmol/L is supported by findings in severe hyperglycemic episodes treated by insulin infusion in oligoanuric patients [21,61,76], in studies of treatment of hyperglycemia in patients with preserved renal function [99,100], and in dogs as well [101]. Extracellular volume abnormalities are a source of [*Na*]*_Cor_* values differing from those computed using the Katz coefficient [21,61]. Formulas calculating the coefficient of the change in [*Na*]*_S_* per the change in [*Glu*]*_S_* at various relationships between the intracellular and extracellular volume have been reported [102,103]. For practical purposes, repeated calculation by the Al-Kudsi formula [16] after each measurement of biochemistry values during the treatment of hyperglycemic emergencies leading to progressive normalization of the extracellular volume will produce progressively more accurate [*Na*]*_Cor_* values [22,25].

Finally, providers should recognize that the method presented in this report addresses only fluid and monovalent cation disturbances. The management of other major pathophysiological disturbances in hyperglycemic emergencies, including the dose and schedule of administration of insulin, the infusion of sodium bicarbonate in DKA, and the replacement of losses in phosphate, magnesium, and other ions, constitutes additional areas of study [76,104].

## 5. Conclusions

The method presented in this report leads to individualization of the volume and composition of the replacement solutions in hyperglycemic emergencies. It can potentially improve their outcomes. However, this method, which increases substantially the complexity of management of these emergencies, has not been applied clinically. Prospective clinical studies are required to assess its performance.

## Figures and Tables

**Figure 1 jcm-14-00025-f001:**
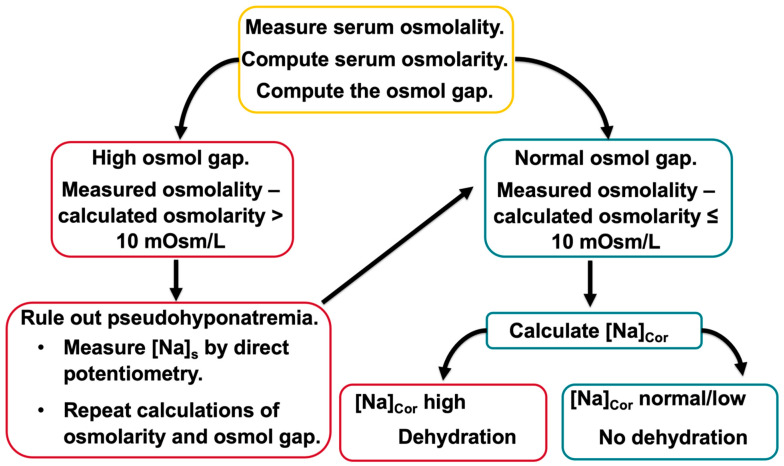
Computing the degree of dehydration in hyperglycemia. Serum osmolality is determined by the total concentration of solutes in the serum water and is expressed in mOsm/kg H_2_O. Serum osmolarity is an expression of total solute concentration in serum, not serum water. It is computed using several formulas, the simplest of which is the following: Osmolarity = 2 × [*Na*]*_S_* + [*Glu*]*_S_* + [*Serum urea*], all in mmol/L. Osmolarity is expressed in mmol/L or mOsm/L. The serum osmol gap is calculated as Measured Osmolality − Computed Osmolarity.

**Figure 2 jcm-14-00025-f002:**
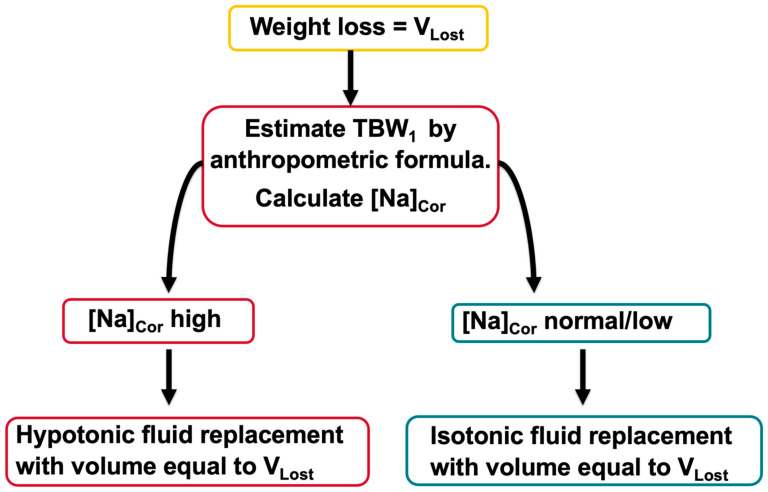
Computing fluid loss in hyperglycemic patients with known weight loss. [*Na*]*_Cor_* should be calculated by Formula (1) as indicated in Figure 1.

**Figure 3 jcm-14-00025-f003:**
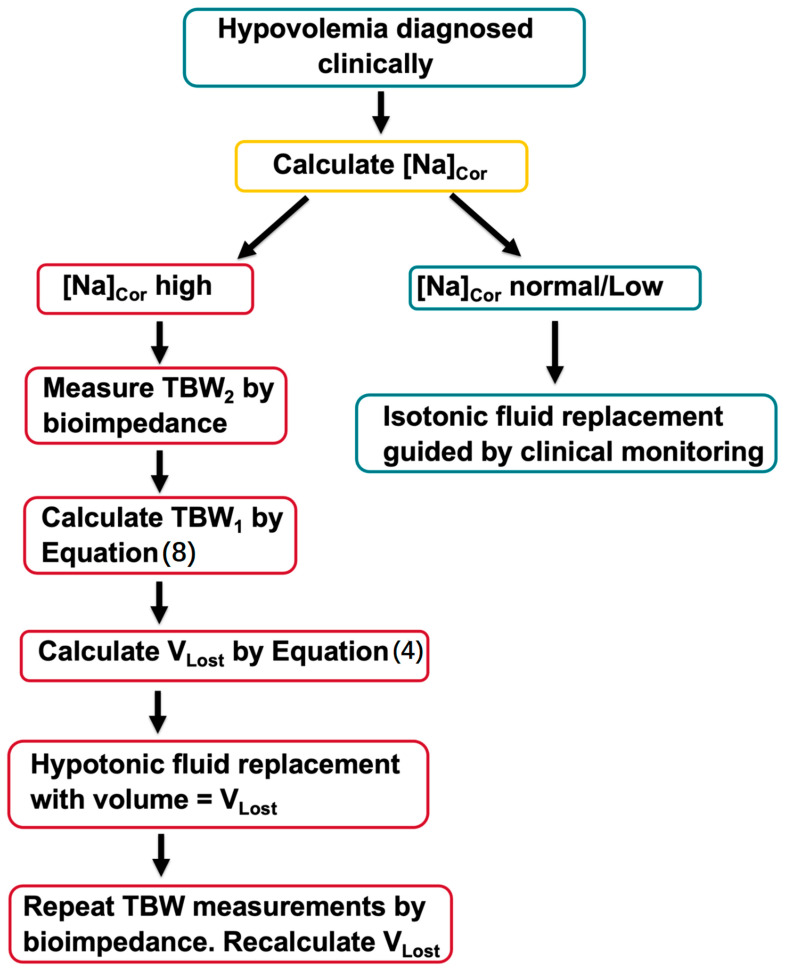
Computing fluid loss in hyperglycemic patients with an unknown weight loss. [*Na*]*_Cor_* should be calculated by Formula (1) as indicated in Figure 1.

**Figure 4 jcm-14-00025-f004:**
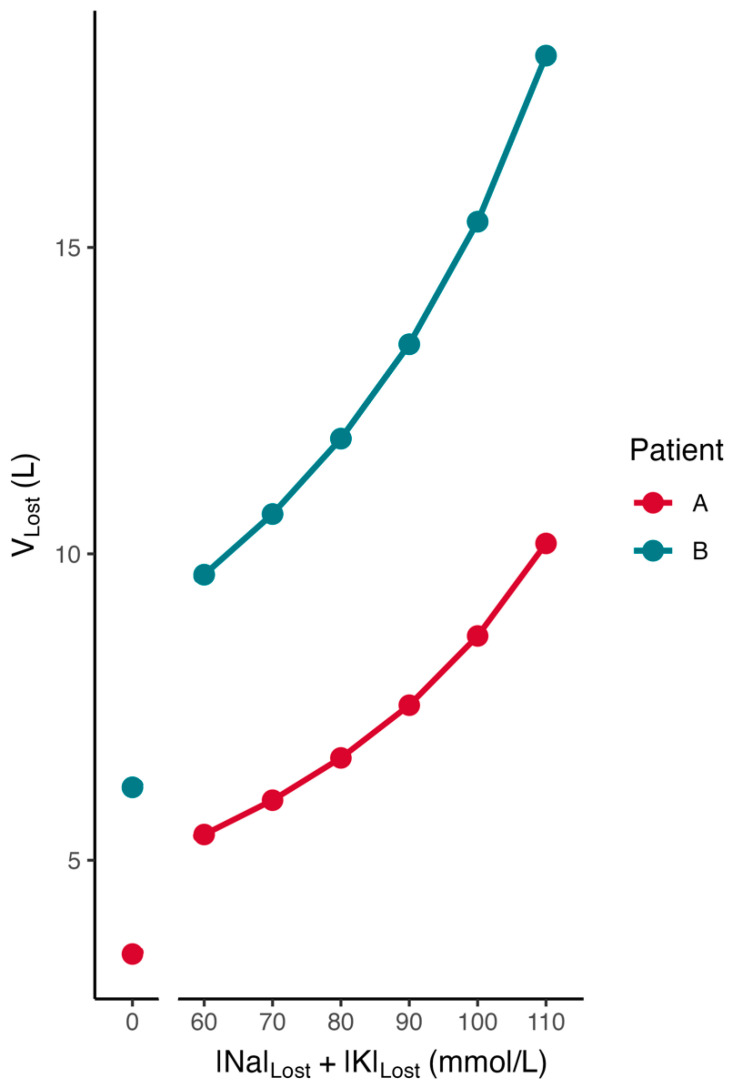
Estimates of the volumes of lost body fluids at different sums of monovalent cation concentrations in two hypothetical patients with the same [*Na*]*_Cor_* but widely differing body water in the baseline hyperglycemic state. *TBW*_1_ was 21.48 L in patient A and 38.27 L in patient B; [*Na*]*_Cor_* was 167.0 mmol/L in both patients.

**Table 1 jcm-14-00025-t001:** Abbreviations used in formulas in this report.

Abbreviation	Explanation
*TBW*	Total body water, *TBW*_1_ in baseline euglycemia, and *TBW*_2_ in hyperglycemia
*V_Lost_*	Volume of fluid lost during the development of hyperglycemia, mostly through osmotic diuresis
[*Glu*]*_S_*	Serum glucose concentration, [*Glu*]*_S_*_1_ in baseline euglycemia (at [*Glu*]*_S_* = 5.6 mmol/L), and [*Glu*]*_S_*_2_ in hyperglycemia
[*Na*]*_S_*	Serum sodium concentration, [*Na*]*_S_*_1_ in baseline euglycemia, and [*Na*]*_S_*_2_ in hyperglycemia
[*Na*]*_Cor_*	Serum sodium concentration corrected for the degree of hyperglycemia
[*Na*]*_Lost_*	Average sodium concentration in the body fluids lost during the development of hyperglycemia
[*K*]*_Lost_*	Average potassium concentration in the body fluids lost during the development of hyperglycemia

Volumes are in L, and glucose and cation concentrations are in mmol/L.

**Table 2 jcm-14-00025-t002:** Formulas estimating the actual losses of body water and monovalent cations and the volumes of body water in the baseline euglycemic state and at presentation with hyperglycemic emergency.

Formula		Reference
$[Na{]}_{Cor}=[Na{]}_{S2}+1.6\times \frac{[Glu{]}_{S2}-5.6}{5.6}$	(1)	[14]
$TBW_{2}=TBW_{1}-V_{Lost}$	(2)	[22]
$V_{Lost}\times ([Na{]}_{Lost}+\left[K{]}_{Lost}\right)=TBW_{1}\times [Na{]}_{S1}-(TBW_{1}-V_{Lost})\times [Na{]}_{Cor}$	(3)	[22]
$V_{Lost}=TBW_{1}\times \frac{[Na{]}_{Cor}-[Na{]}_{S1}}{\left[Na{]}_{Cor}-([Na{]}_{Lost}+[K{]}_{Lost}\right)}\;$	(4)	[22]
$[Na{]}_{Lost}+[K{]}_{Lost}=\frac{TBW_{1}\times [Na{]}_{S1}-(TBW_{1}-V_{Lost})\times [Na{]}_{Cor}}{V_{Lost}}$	(5)	
$TBW_{2}=TBW_{1}\times (1-\frac{[Na{]}_{Cor}-[Na{]}_{S1}}{\left[Na{]}_{Cor}-([Na{]}_{Lost}+[K{]}_{Lost}\right)})$	(6)	
$TBW_{1}=V_{Lost}\times \frac{\left[Na{]}_{Cor}-([Na{]}_{Lost}+[K{]}_{Lost}\right)}{[Na{]}_{Cor}-[Na{]}_{S1}}$	(7)	
$TBW_{1}=\frac{TBW_{2}}{1-\frac{[Na{]}_{Cor}-[Na{]}_{S1}}{\left[Na{]}_{Cor}-([Na{]}_{Lost}+[K{]}_{Lost}\right)}}\;$	(8)	

**Table 3 jcm-14-00025-t003:** Case of DKA-HHS. Laboratory and computed values on admission.

Parameter	Value	Measurement/Computation
Height, cm	160	Admission
Weight_1_, kg	46.6	10 days prior to admission
Weight_2_, kg	36.8	Admission
[*Glu*]*_S_*, mmol/L	123.7	Admission
[*Na*]*_S_*, mmol/L	135	Admission
*TBW*_1_, L	30.4	−21.993 + 0.209 × 160 + 0.406 × 46.6—Ref. [58]
*V_Lost_*, L	9.8	46.6 − 36.8
[*Na*]*_Cor_*, mmol/L	168.7	135 + 1.6 × (123.7 − 5.6)/5.6, Formula (1)
[*Na*]*_Lost_* + [*K*]*_Lost_*, mmol/L	80	{30.4 × 140 − (30.4 − 9.8) × 168.7}/9.8, Formula (5)

Note: From Formula (2), *TBW*_2_ = 30.4 − 9.8 = 20.6 L; from Formula (6), *TBW*_2_ = 30.4 × {1 − (168.7 − 140)/(168.7 − 80)} = 20.6 L.

**Table 4 jcm-14-00025-t004:** Computation of the mean fractional water loss in the Arieff and Carroll study.

Parameter	Value	Computation
[*Glu*]*_S_*, mmol/L	64.8	
[*Na*]*_S_*, mmol/L	144	
[*Na*]*_Cor_*, mmol/L	160.9	144 + 1.6 × (64.8 − 5.6)/5.6, Formula (1)
*V_Lost_*, L	9.1	Volume of fluid retained during treatment
Na_1_, K_2_ gain, mmol	407_1_, 137_2_	Retained during treatment
[*Na*]*_Lost_* + [*K*]*_Lost_*, mmol	59.8	407/9.1 + 137/9.1
*TBW*_1_, L	44	9.1 × (160.9 − 59.8)/(160.9 − 140), Formula (7)
*V_Lost_*/*TBW*_1_	0.207	9.1/44

_1_: amount of sodium retained. _2_: amount of potassium retained.

**Table 5 jcm-14-00025-t005:** Mean or median serum glucose, sodium, and corrected sodium reported in studies of patients with hypernatremia during hyperglycemic crises.

[*Glu*]*_S_* mmol/L	[*Glu*]*_S_* mg/dL	[*Na*]*_S_*_2_ mmol/L	[*Na*]*_Cor_* mmol/L	Number of Cases	Reference
49.7	894.6	160.0	173.6	155	[22]
40.7	732.6	153.0	163.1	132	[33]
56.1	1009.7	153.4	168.0	21	[66]
52.2	940.3	153.0	167.0	6	[67]
51.4	926.0	145.9	159.1	12	[68]
59.4	1079.0	148.0	163.7	23	[69]
35.6	640.8	148.8	157.5	41	[70]

**Table 6 jcm-14-00025-t006:** Total body water at the baseline euglycemic state and volume of lost water computed using different *TBW*_2_ and values of [*Na*]*_Lost_* + [*K*]*_Lost_* in hypothetical patients with hyperglycemia.

Patient	[*Na*]*_Cor_* nmol/L	*TBW*_2_ L	[*Na*]*_Lost_* + [*K*]*_Lost_* mmol/L	*TBW*_1_ L	*V_Lost_* L	*V_Lost_*/*TBW*_1_
1	173.6	20	0	24.80	4.80	0.1935
2	173.6	40	0	49.60	9.60	0.1935
3	173.6	20	60	28.40	8.10	0.2958
4	173.6	40	60	56.80	16.80	0.2958
5	157.5	20	0	22.50	2.50	0.1111
6	157.5	40	0	45.00	5.00	0.1111
7	157.5	20	60	24.38	4.38	0.1795
8	157.5	40	60	48.76	8.76	0.1795
9	157.5	40	110	63.33	23.33	0.3684

## Data Availability

Data are contained within the article.

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
