# Peer review of "Quantifying the Deficits of Body Water and Monovalent Cations in Hyperglycemic Emergencies"

_jcm, 2024, doi:10.3390/jcm14010025_

Round 1

Reviewer 1 Report

Comments and Suggestions for Authors

This manuscript presents a well-reasoned and justified approach to estimating free water and cation loss in diabetic ketoacidosis. While the reasoning and approach seems sound, their results do not convincingly demonstrate that this approach is valid or necessary in real patients, which I think is critical. This approach substantially increases the complexity of DKA management above currently recommended approaches, so stronger results are needed. 

Comments

1.     The “Results” section is lacking. Only a single real patient is presented; whether this patient’s poor outcome was due to inadequate volume resuscitation (as suggested by their corrected Na levels) or by failure to respond to the patient’s large diuresis is not clear. Addition of other real patients (either prospective or retrospective) that includes how much their estimates differ from standard approaches would be helpful to demonstrate whether their approach may be helpful in fluid management. 

2.     The second example of Group A results presents the mean of aggregate results from a prior publication. Since there is wide variability between patients (which helps justify the need for such a personalized approach that they are proposing), it would be much stronger to present calculations on individual patients, with relevant variability reporting, rather than calculations on a population mean. 

3.     The group B results are all on “hypothetical” patients. This data is important for establishing the values to use in the method, but validation in real patients is critical.  

4.     When presenting patients, it would be helpful to include a table with key clinical characteristics of the presented patients, as is generally included in clinical trial articles. 

5.     The paper, though clearly written, is very long for the stated purpose (presenting a new method to estimate body water loss and electrolyte loss in DKA). I’d recommend narrowing the focus by decreasing the amount of information provided about identifying precipitants of DKA, clinical characteristics of DKA, and ruling out pseudohypernatremia, mentioning these feature and providing references the reader can go to for these basic points. 

6.     The Discussion section is also quite long for the amount of data presented. Narrowing and focusing the discussion would be helpful. 

7.     The authors appropriately acknowledge significant limitations of their study, but an important limitation they don’t touch on is the clinical feasibility of the approach. This approach not only requires bioimpedance measurement (which they acknowledge), but also requires significantly more calculations, custom IV fluid preparation, and ongoing urine monitoring (in addition to standard practice of serum monitoring). 

Reviewer 2 Report

Comments and Suggestions for Authors

This is a very interesting proof-of-concept manuscript describing the development of formulas that will aid in individualized approach to corrections of hypovolemia in hyperglicemic emergencies. The authors are commended for the clarity of presentation and consideration of confounding factors that may apply in these situations. Please find below a few points that in my opinion will improve the manuscript:

1. In the computation of losses section, you list and describe the information necessary for the computation of losses. I haven't seen a discussion of the approach to  a patient that presents comatose and for which crucial bit of information , like underlying ESRD cannot be obtained. Could the author either provide this as a limitation or describe an approach to this clinical scenario?

2. In the description of the approach to the boy with DKA, it is unclear as to what advantage would the use of the derived formulas provide? The case culminates in the boy losing consciousness due to too rapid of a sodium correction. Could the author perhaps clarify what went wrong and how the approach they are describing in this manuscript could have helped? Perhaps it is just a matter of clarifying, I had a hard drawing these connections. 
